# The Impact of Processing Parameters on the Content of Phenolic Compounds in New Gluten-Free Precooked Buckwheat Pasta

**DOI:** 10.3390/molecules24071262

**Published:** 2019-04-01

**Authors:** Anna Oniszczuk, Kamila Kasprzak, Agnieszka Wójtowicz, Tomasz Oniszczuk, Marta Olech

**Affiliations:** 1Department of Inorganic Chemistry, Medical University of Lublin, ul. Chodźki 4a, 20-093 Lublin, Poland; kasprzak.kamila.k@o2.pl; 2Department of Thermal Technology and Food Process Engineering, University of Life Sciences in Lublin, ul. Głęboka 31, 20-612 Lublin, Poland; agnieszka.wojtowicz@up.lublin.pl; 3Department of Pharmaceutical Botany, Medical University of Lublin, ul. Chodźki 1, 20-093 Lublin, Poland; martaolech@umlub.pl

**Keywords:** liquid chromatography, functional food, gluten-free pasta, buckwheat, antioxidant activity, phenolic acids, extrusion-cooking, processing parameters

## Abstract

Buckwheat is a generous source of phenolic compounds, vitamins and essential amino acids. This paper discusses the procedure of obtaining innovative gluten-free, precooked pastas from roasted buckwheat grains flour, a fertile source of natural antioxidants, among them, phenolic acids. The authors also determined the effect of the extruder screw speed and the level of moisture content in the raw material on the quantity of free phenolic acids. The qualitative and quantitative analysis of phenolic acids in pasta was carried out using high-performance liquid chromatography electrospray ionization tandem mass spectrometry (HPLC-ESI-MS/MS). The chromatographic method was validated. For extracts with the highest total content of free phenolic acids and unprocessed flour from roasted buckwheat grain, the TLC-DPPH test was also performed to determine the antioxidant properties of the tested pasta. The level of moisture in the raw material had an impact on the content of phenolic acids. All pastas made from buckwheat flour moistened up to 32% exhibited a higher total content of free phenolic acids than other mixes moistened to 30 and 34% of water.

## 1. Introduction

In recent years, there have been attempts at neutralizing the negative influence of free radicals on the human body. Reactive species contribute to the development of numerous disorders, including cancers and neurodegenerative diseases. Free radical scavenging ability is one of the most desirable features of natural products. In a vast majority of cases, plant extracts abound in valuable antioxidants, such as phenolic acids, flavonoids, or carotenoids. Much attention is attached to substances contained in raw fruit and vegetables and relatively less to industrial treated plants used for food production, e.g., noodles, crisps, or breakfast cereals. This approach to antioxidants and their content in food has a pivotal role in our daily nutrition and combating free radicals.

Among functional foods, buckwheat is cultivated and consumed worldwide. Common and Tartary buckwheat (*Fagopyrum esculentum* Moench and *F. tataricum* Gaertner, respectively) are commonly consumed [1]. Several studies in the field of functional food indicated a high content of numerous vitamins, antioxidants and well-balanced amino acid compositions in the plant [2,3]. Studies carried out towards identification of free radical scavengers revealed a high content of phenolic acids, rutin, quercetin, isoquercetin, kaemferol and many other substances in different parts of the plant [4], with seeds confirmed to be the most valuable as functional foods. Yao et al. [5] and Wang et al. [6] revealed a positive influence of the Tartary buckwheat on the reduction of the glucose level in blood and adjustment of the lipid profile. Additionally, some scientific studies also exposed some other properties of the plant, just to mention antibacterial, anti-inflammatory, and antidiabetic [4]. Due to the high content of antioxidants, numerous minerals and nutritional components, buckwheat is cultivated for seeds used in the food industry. The importance of active compounds in human health has stimulated a research on functional foods and plants, including buckwheat. 

A convenient and practical method of production of foodstuffs based on or with the addition of buckwheat is extrusion-cooking. This process involves a mechanical and thermal treatment of raw material under high pressure. This processing method combines a sequence of mixing, heating, flow, forming, and shaping. Extruder machines end with a forming die. Any water contained in the processed material—due to its cooking under the treatment conditions—evaporates quickly as steam when it leaves the forming die, which causes the expansion of the material and the formation of its porous structure [7]. In the case of instant or precooked pasta, the expansion phase must be eliminated, and the final section of the extruder plasticizing unit must be cooled down so that the temperature of the product exiting the die does not exceed 100 °C as it prevents the formation of the porous structure and reduces pasta stickiness [8]. Extruded pasta, both from conventional (durum wheat and common wheat) and from gluten-free raw materials (rice, maize, legume seeds) exhibit a high degree of gelatinization of starch and a dense internal structure without empty spaces inside the pasta cross-section [7,8,9]. 

The choice of extrusion-cooking parameters is of utmost importance because this high-temperature/high-pressure process may, if handled improperly, cause the destruction of thermolabile active compounds in the product [10,11]. So far, no research has looked into the impact of production parameters on the content of phenolic acids in extruded precooked pasta products made from flour from roasted buckwheat grain. Hence, the objective of our experiment is to test the effect of the rotational speed of the extruder screw and the growing level of moisture content in the raw material on the quantity of free phenolic acids in the said product.

## 2. Results and Discussion

### 2.1. Quantitative Analysis of Phenolic Acids (HPLC-ESI-MS/MS)

The qualitative and quantitative analysis of phenolic acids in extruded pasta made from roasted buckwheat flour was carried out using high-performance liquid chromatography electrospray ionization tandem mass spectrometry (HPLC-ESI-MS/MS). The chromatographic method was validated. Calibration curve equations, limit of detection (LOD) and limit of quantification (LOQ) and the ranges of linearity for assays of individual phenolic acids are shown in Table 1. 

The extraction method used was 40-min ultrasonic-assisted extraction at elevated temperature (60 °C) with the use of an 80% aqueous ethanol solution. This method was selected because in previous experiments it had proven to be the optimum technique for the isolation of phenolic acids from buckwheat pastas [12]. 

Buckwheat pastas were produced at different rotational speeds of the extruder screw (60, 80, 100 and 120 rpm) and different levels of raw material moisture (30, 32 and 34%). The following phenolic acids were detected in obtained samples: gallic, protocatechuic, gentisic, 4-hydroxybenzoic, vanilic, *trans-*caffeic, *cis*-caffeic, *trans*-*p*-coumaric, *cis-p*-coumaric, syryngic, *trans*-ferulic, *cis*-ferulic, salicylic, *trans-*sinapic and *cis*-sinapic (Appendix A, Table 2). These compounds were present in all pastas, regardless of moisture content and screw speed. The presence of syryngic, *trans*-ferulic and sinapic acids in the pasta was lower than the limit of quantification but higher than the limit of detection. 

The highest amounts of the vast majority of benzoic acid derivatives, i.e., gallic, protocatechuic, gentisic, 4-hydroxybenzoic and salicylic acids, were reported in samples produced at 100 rpm. of the extruder screw and flour moisture of 32%. However, the highest content of cinnamic acid derivatives —*trans*-caffeic, *trans-p*-coumaric, *cis*-*p*-coumaric and *cis*-ferulic was observed for buckwheat pasta produced with 30% of flour moisture and 60 rpm of the extruder screw. Besides, it should be noted that, in roasted buckwheat flour, very small quantities of all cinnamic acid derivatives are present, with the exception of *trans-p*-coumaric acid, even lower than their limits of quantification.

As noted by Zieliński et al. [13], phenolic acids in plants occur mainly in a bound form as the components of lignins and hydrolysing tannins and in the form of esters and glycosides. Some of hydroxycinnamic acids are found in ester combinations with carboxylic acids or glucose, while hydroxybenzoic acids are mostly present as glycosides. In grain kernels, ferulic and p-coumaric acids are mostly bound with arabinoxylans. Moreover, plant tissues reveal combinations of phenolic acids with other natural compounds, e.g., flavonoids, fatty acids, or sterols. 

In buckwheat, however, most of benzoic acid derivatives are present in free form [14,15]. These compounds are found across the entire grain. On the other hand, cinnamic acid derivatives, in particular ferulic and p-coumaric acids, are found in the primary cell wall, especially in hemicelluloses. Our test results are by far compatible with the theory outlined above: benzoic acid derivatives are present both in unprocessed flour from roasted buckwheat and in buckwheat pasta, and the extruder screw speed of 100 rpm. with 32% moisture content of the mixture creates optimum conditions to release hydroxybenzoic acids from the few combinations that they can create with other compounds (e.g., glycosidic) without deactivating aglycons. In both flour and pasta, the prevailing acids were: gallic, protocatechuic, and 4-hydroxybenzoic.

Hydroxycinnamic acids, which cannot be quantified in unprocessed buckwheat flour, appear in pasta, and the use of the extruder screw speed of 60 rpm at 30% moisture content of the raw material creates optimum conditions (at least, of all tested conditions) to release *trans-p*-coumaric, *cis*-*p*-coumaric and *cis*-ferulic acids from combinations with other compounds. Similar test results were reported by Zieliński et al. [13]. They showed that ferulic acid content in cereal grains increases even five-fold after the extrusion-cooking process. Hydrothermal treatment, e.g., extrusion at extremely high temperatures, is likely to lead to reduced digestibility or to the formation of toxic or mutagenic compounds, e.g., Maillard reaction products, or to the formation of acrylamide in extruded products [16]. 

On the other hand, such compounds can be formed that display health properties and have a beneficial effect on the human body [17,18]. Increasingly popular in food technology and employed for the processing of cereal or pseudocereal material, the extrusion-cooking process makes the final product exhibit diversified properties. The action of temperature, pressure and shear forces on moist raw material induces profound changes in the processed matter in a very short time, among them: enhanced digestibility of nutrients, inactivation of anti-nutritive factors, modified sensory characteristics. The intensity of these changes depends both on the properties of raw material (e.g., humidity) and on the settings of the extrusion-cooking procedure, involving the temperature or rotational speed of the extruder screw. A high degree of mixing and homogenisation leads to a decrease in diffusion barriers and the breaking down of chemical bonds, which results in a heightened reactivity of the components. Properly selected extrusion-cooking conditions may release phenolic acids from the chemical bonds that they create with other compounds (e.g., glycosidic bonds) without deactivating aglycones [9,19]. According to the research done by Alonso et al. [20], the main factors stimulating the transformation of input material during the extrusion-cooking process are high temperatures and mechanical aspects related to shear forces, which occur along with the increase of screw rotational speed. Therefore, the proper selection of manufacturing parameters is paramount. In the studied case, humidity proved to be the determining factor. However, no apparent tendency was observed as regards the impact of the rotational speed of the extruder screw on the content of phenolic acids. At 30% and 34% of moisture content, the largest quantity of phenolic acids was seen in buckwheat pasta produced at the screw speed of 60 rpm, while at 32%–in products extruded at the screw speed of 100 rpm. In general, regardless of the screw rotational speed, all the pastas made from the raw material moistened up to 32% showed a higher total content of free phenolic acids than when the content of moisture was 30 and 34%. The highest total content of free phenolic acids (12,454 μg/g s.m.) was reported in pasta produced at 100 rpm and flour moisture at 32%. 

### 2.2. Antioxidant Properties of Gluten-Free Buckwheat Pasta

It is common property that aglycones show a higher antioxidant activity than glycosidic forms or ones bound by other types of bonds [21]. The antioxidant activity of phenolic acids also depends on the number of hydroxyl groups in the molecule and can be augmented by spherical effects. Cinnamic acid derivatives are more effective antioxidants than benzoic acid derivatives [21,22]. Phenolic acids are responsible for free radical quenching, protection of lipids and proteins against peroxidation as well as having the ability to chelate transition metal ions that catalyze oxidation reactions [23]. Therefore, the next stage of the research covered a TLC-DPPH test (thin layer chromatography-2,2-diphenyl-1-picrylhydrazyl test) of extracts with the highest total content of free phenolic acids obtained from buckwheat pastas made from roasted buckwheat flour moistened up to 32%. This experiment was aimed at assessing the antioxidant properties of the pasta produced. After applying, developing and drying the extracts, the plate was sprayed with DPPH (2,2-diphenyl-1-picrylhydrazyl) and then scanned after 0, 5, 10, 15 and 30 min. Images saved as JPGs were processed into numerical data in the Sorbfil TLC Videodensitometer application. The function of result analysis permitted a precise assessment of the colour intensity of the spots and the size of the areas under peaks (and their conversion into numerical data). The area of a standard solution of rutin at the concentration of 0.1 mg/mL was adopted as a reference point. Its activity was marked “1” [24]. 

The TLC-DPPH test showed that the pastas made after moisturizing flour up to 32% demonstrate antioxidant properties similar to 0.1mg/mL rutin solution (Table 3, Figure 1). By far, the highest DPPH free radical scavenging activity was seen in the flour from roasted buckwheat, and, among the tested pasta, in products obtained at the extruder screw speed of 100 rpm. The other three samples showed a similar activity towards free radical DPPH. It was demonstrated that samples with the highest total content of free phenolic acids exhibited the highest antiradical activity. Buckwheat flour, for which the highest activity was reported, only underwent toasting. Next to free phenolic acids, it obviously contains several compounds that are relevant as antioxidants originating naturally in kernel. However, in the case of extruded pasta, although the content of free phenolic acids is by and large the same as in flour, the antioxidant properties are markedly lower. The extrusion-cooking process may be responsible for that. The integrated effect of temperature, pressure and shear forces that occurred during the extrusion-cooking enhances digestibility and ability to extract phenolic acid from products, yet it can also reduce the amount of other antioxidant compounds (e.g., glutathione, inositol, melatonin, tocopherols). Therefore, the selection of proper production parameters is key when developing pasta that should exhibit the most beneficial composition and content of plant active compounds for the human body.

## 3. Materials and Methods 

### 3.1. Preparation of Pasta

The raw material for the preparation of buckwheat pasta was flour obtained by grinding dehulled and roasted buckwheat grains. The chemical composition of the target flour was determined according to the American Association of Cereal Chemists Approved Methods (AACC 2000): protein 11.39%, fat 1.06%, ash 1.65%, fiber 9.3%. The raw material was moistened with a proper amount of water and mixed to 30, 32 and 34% of moisture content in the dough. After mixing, the raw materials were processed in a modified TS-45 single-screw extruder (Metalchem, Gliwice, Poland) with a plasticizing system configuration of L/D = 18:1 and compression ratio 3:1; the extruder was equipped with an additional glycol cooling section (SW MINI 8P cooler from Cool, Chotomów, Poland) in the closing section of the cylinder; the temperature range of the extruder was from 90 to 110 °C in the plasticizing section and from 55 to 65 °C in the cooling section. The pasta was shaped on a forming die with 12 round openings with a diameter of 0.8 mm each. It was produced at various screw speeds, i.e., 60, 80, 100 and 120 rpm. After drying at a temperature below 40 °C in an air oven overnight, the samples were stored in closed containers.

### 3.2. Preparation Of Polyphenolic Extracts

Ultrasound-assisted extraction (UAE) was used to obtain extracts. The extraction temperature and time, as well as the type and amount of extracting solvent, were optimized in advance [12]. 2 g of samples were prepared and transferred quantitatively to ground-necked flasks. 80% ethanol was added. The extraction-cooking procedure was carried out in an ultrasonic bath (20 kHz, 100 W; BANDELIN electronic GmbH & Co. KG, Berlin, Germany) at 60 °C in two 20-min cycles for each batch of the material. 40 mL of methanol was added to each batch. The obtained extracts were combined, filtered through a pleated paper filter, and then the solvent was evaporated to a total volume of 5 mL for each sample. Before the analysis, the extracts were filtered through a syringe filter of 0.45 μm of the pore size. Each extraction was repeated three times.

### 3.3. Quantitative Analysis of Phenolic Acids Using High-Performance Liquid Chromatography Electrospray Ionization Tandem Mass Spectrometry (HPLC-ESI-MS/MS)

The qualitative and quantitative analysis of phenolic acid content was carried out using high-performance liquid chromatography electrospray ionization tandem mass spectrometry (HPLC-ESI-MS/MS) [12]. An Agilent 1200 Series liquid chromatograph (Agilent Technologies, Santa Clara, CA, USA) was used in the tests. Chromatographic separation was carried out on a Zorbax SB-C18 column (2.1 × 50 mm, 1.8 μm, Agilent Technologies) at 25 °C. Mobile phases were used as follows: 0.1% (*v*/*v*) aqueous formic acid solution (A) and 0.1% (*v*/*v*) formic acid solution in methanol (B). The analysis was carried out in a gradient layout at a constant eluent flow of 400 μL/min and the following gradient composition: 0–1 min: 95% A and 5% B; 2–4 min: 80% A and 20% B; 8–9.5 min: 30% A and 70%; 11.5–15 min: 95% A and 5% B, three times for each sample. The volume of the sample applied was 3 μL. 

Detection was carried out on a triple quadrupole ion trap mass spectrometer QTRAP 3200 (AB Sciex, Framingham, MA, USA). Ionization was administered by spraying in the electric field in the negative mode ESI (−) (electrospray ionization); the following mass detector parameters were applied: ion source voltage—4500 V, capillary temperature 400 °C, shielding gas 30 psi, spraying gas 60 psi. The examination of phenolic acid fragmentation was carried out in the *m*/*z* range from 50 to 400. Identification and determination of phenolic acids in the plant material was done by monitoring of selected reactions of metastable ion formation (Multiple Reaction Monitoring—MRM). External standard method was used for quantification. For all analytes linearity range of calibration curve, limit of detection (LOD) and limit of quantification (LOQ) were determined. 

### 3.4. TLC-DPPH Test 

The antioxidant properties of extracts with the highest content of free phenolic acids were assessed with the TLC-DPPH test. Silica gel plates were used as the stationary phase. A mixture of acetonitrile, water, chloroform, formic acid (30:2:5:2, *v*/*v*/*v*/*v*) was used as the mobile phase. The extracts along with a reference solution of rutin at the concentration of 0.1 mg/mL were applied using an automatic TLC applicator (Desaga AS-30, Hamburg, Germany). After development in a horizontal chamber, the plate was dried and sprayed with a 0.1% DPPH methanol solution. Next, it was scanned after 0, 5, 10, 15 and 30 min. The Sorbfil TLC Videodensitometrs computer program was used to convert obtained results into numerical data [23].

## Figures and Tables

**Figure 1 molecules-24-01262-f001:**
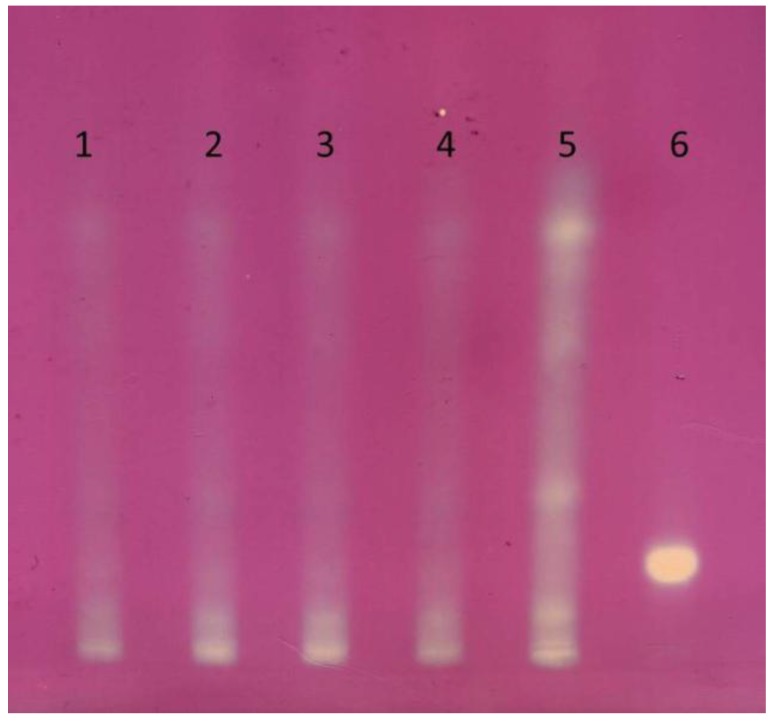
Results of TLC-DPPH; scan after 30 min. Mobile phase: acetonitrile, water, chloroform, formic acid (30:2:5:2, *v*/*v*/*v*/*v*). 1—pasta 32/60, 2—pasta 32/80, 3—pasta 32/100, 4—pasta 32/120, 5—roasted buckwheat flour, 6—rutin (activity of rutin solution equal to “1”, see Table 3).

**Table 1 molecules-24-01262-t001:** Analytical parameters of LC-MS/MS quantitative method; data for calibration curves, limit of detection (LOD) and limit of quantification (LOQ) values for each analyzed phenolic acid.

Compound	Regression Equation	LOD [ng/mL]	LOQ [ng/mL]	*r* ^2^	Linearity Range [ng/mL]
gallic	y = 214x + 808	25	50	0.9992	50–5000
protocatechuic	y = 86.3x + 1240	10	25	0.9999	50–12500
gentisic	y = 717x + 45100	5	15	0.9994	15–5000
4-OH-benzoic	y = 1470x + 7020	20	50	0.9999	50–2000
vanillic	y = 59x + 1950	25	50	0.9995	50–5000
*trans*-caffeic	y = 1080x + 6640	10	25	0.9992	25–2500
*cis*-caffeic	y = 1080x + 6640	10	25	0.9992	25–2500
syryngic	y = 70.9x + 8720	25	50	0.9993	50–10000
*trans*-*p*-coumaric	y = 888x + 855	10	25	0.9999	25–2500
*cis*-*p*-coumaric	y = 888x + 855	10	25	0.9999	25–2500
*trans*-ferulic	y = 360x − 3740	10	25	0.9994	40–2000
*cis*-ferulic	y = 360x − 3740	10	25	0.9994	40–2000
*trans*-synapic	y = 302x + 2570	8	20	0.9996	20–2500
*cis*-synapic	y = 302x + 2570	8	20	0.9996	20–2500
salicylic	y = 2060x + 13000	10	30	0.9999	30–1000

**Table 2 molecules-24-01262-t002:** Content of phenolic acids in gluten-free precooked buckwheat pasta (n = 3, mean ± SD).

MC (%)	SS (rpm)	Content of Phenolic Acids (µg/g)											
		Gallic	Protocatechuic	Gentisic	4-OH-Benzoic	Vanilic	*trans*-Caffeic	*cis*-Caffeic	*trans*-*p*-Coumaric	*cis*-*p*-Coumaric	*cis*-Ferulic	Salicylic	Sum
RBF		3.185	4.120	0.167	2.305	0.659	BQL	BQL	0.304	BQL	BQL	2.460	13.200
SD		0.0212	0.2828	0.0035	0.0071	0.0007	-	-	0.0070	-	-	0.0014	
30	60	2.215	2.240	0.283	2.100	0.651	0.182	0.063	0.652	0.357	0.472	0.956	10.189
		0.0353	0.0014	0.0014	0.0141	0.0007	0.0035	0.0002	0.0084	0.0014	0.0007	0.0021	
	80	2.570	2.585	0.320	2.080	0.495	0.133	0.071	0.472	0.239	0.256	0.926	10.147
		0.0141	0.0212	0.0028	0.0014	0.0071	0.000	0.0003	0.0014	0.0070	0.0018	0.0070	
	100	1.972	1.938	0.219	1.440	0.309	0.103	0.049	0.216	0.155	0.148	0.435	7.029
		0.0000	0.0141	0.0001	0.0014	0.0017	0.000-	0.000	0.0042	0.0028	0.0012	0.0035	
	120	2.190	2.030	0.287	2.060	0.425	0.130	0.080	0.354	0.191	0.153	0.813	8.713
		0.0141	0.0424	0.0021	0.0014	0.0012	0.0014	0.0003	0.0099	0.00141	0.0007	0,0000	
32	60	2.760	2.765	0.314	3.065	0.388	0.124	0.061	0.357	0.204	0.129	0.904	11.071
		0.0565	0.0070	0.0021	0.0070	0.0035	0.0007	0.0005	0.0042	0.0042	0.0012	0.0014	
	80	2.465	2.670	0.329	2.765	0.326	0.134	0.100	0.361	0.256	0.195	0.932	10.533
		0.0212	0.0141	0.0035	0.0070	0.0049	0.0007	0.0006	0.0063	0.0056	0.0023	0.0042	
	100	3.115	3.105	0.414	3.150	0.351	0.157	0.122	0.395	0.309	0.197	1.140	12.455
		0.0035	0.0070	0.0042	0.0014	0.0021	0.0001	0.0007	0.0085	0.0021	0.0014	0.0141	
	120	2.595	2.755	0.340	2.835	0.456	0.155	0.060	0.419	0.205	0.118	0.953	10.911
		0.0070	0.0212	0.0212	0.2121	0.0530	0.0007	0.0001	0.0021	0.0014	0.0013	0.0049	
34	60	2.585	2.710	0.353	2.290	0.443	0.111	0.054	0.347	0.299	0.130	0.932	10.254
		0.007	0.0003	0.0021	0.0011	0.0088	0.0014	0.0001	0.0014	0.0021	0.0007	0.0035	
	80	1.915	2.195	0.222	2.335	0.379	0.119	0.036	0.299	0.211	0.124	0.675	8.510
		0.0070	0.0071	0.0049	0.0054	0.0006	0.0001	0.0002	0.0056	0.0021	0.0003	0.0028	
	100	2.240	2.310	0.304	2.180	0.377	0.155	0.040	0.397	0.186	0.128	0.799	9.116
		0.0056	0.0282	0.0106	0.0141	0.0021	0.0035	0.0001	0.0077	0.0014	0.0008	0.0007	
	120	2.080	2.025	0.298	1.930	0.307	0.114	0.058	0.300	0.266	0.125	0.784	8.287
		0.0014	0.0210	0.0035	0.0000	0.0028	0.0042	0.0006	0.0028	0.0007	0.0049	0.0035	

MC—moisture content, SS—screw speed, RBF—roasted buckwheat flour, SD—standard deviation, BQL—results above LOD and below LOQ.

**Table 3 molecules-24-01262-t003:** Results of TLC-DPPH assay showing the antiradical activity of analyzed extracts in relation to the activity of 0.1 mg/mL rutin solution (activity of rutin solution equal to “1”).

Time	Pasta 32/60 1	Pasta 32/80 2	Pasta 32/100 3	Pasta 32/120 4	Roasted Buckwheat Flour 5
0 min	0.788	0.831	0.717	0.780	2.095
5 min	0.806	0.886	0.772	0.835	2.278
10 min	1.049	1.018	1.141	0.979	2.318
15 min	1.009	1.021	1.277	0.847	2.251
30 min	1.116	1.194	1.368	1.095	2.472

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
