# Peer review of "The Impact of Processing Parameters on the Content of Phenolic Compounds in New Gluten-Free Precooked Buckwheat Pasta"

_molecules, 2019, doi:10.3390/molecules24071262_

Round 1

Reviewer 1 Report

The science is sufficient but this manuscript is very 'wordy'. The verbiage needs to be condensed; many of the sentences can be more succinctly written without affecting what the authors are trying to convey. Following are examples:

Line 15: Change, "The research covered in the paper aimed..." to, "..this paper aims..."

Line 17-19: "...The effect of the rotational speed of the extruder screw and the level of moisture content in the raw material on the quantity of free phenolic acids was also determined."

Line 19: Delete ...tested...

Line 23: Change, "...make a preliminary assessment of..." to," ...determine the..."

Line 24: Delete ...tested...

Line 24-25: Change, "...proved to be the determining factor as regards..." to, "...affected..."

Line 26: % of what???

Line 39-40: Change, "Among available functional foods, buckwheat has a special place in daily nutrition. It is cultivated in many parts of the world." to, "Among functional foods, buckwheat is cultivated and consumed worldwide."

Line 40-42: Change to, Common and Tartary buckwheat (Fagopyrum esculentum Moench and F. tataricum Gaertner, respectively) are commonly consumed [1]

Author Response

Dear Reviewer,

The authors would like to thank the Reviewer for valuable comments. Manuscript have been checked by a native English speaking person from “Two Heads. Coaching and Language” company. The proper changes have been introduced.

Yours sincerely,

Anna Oniszczuk

Reviewer 2 Report

                                     Comments

       This manuscript described the processing parameters (e.g., the rotational speed of the extruder screw and levels of raw material moisture) on the content of phenolic compounds in gluten-free pre-cooked buckwheat pasta, and compared the antioxidant properties of the tested pasta with the flour from roasted buckwheat. This is an interesting study that is more related to food technology. However, for an academic paper or scientific study, the detailed scientific problem is less involved in this manuscript. The studied problem in the current manuscript only belongs the field of food technology rather than a problem in food science. In my opinion, the current manuscript is not adequate for publication in Molecules.

The questions and suggestions are listed as follows:

1)    In the whole manuscript, the mechanism of the effects of processing parameters on the content of phenolic compounds is not analyzed and discussed.

2)    This research received no external funding, just in my opinion, I consider that this study is not a scientific research in the current situation, but a technical study.

3)    HPLC-ESI-MS/MS means “high-performance liquid chromatography electrospray ionization tandem mass spectrometry” rather than “high performance liquid chromatography coupled with mass spectrometry”, the authors should revise it through the whole manuscript.

4)    A total of 15 phenolic acids described, why only 11 phenolic acids presented in Table 2?

5)    Page 3, line 94-96. The authors said that “However, the highest content of cinnamic acid derivatives trans-caffeic, trans-p-coumaric, cis-p-coumaric and cis-ferulic was observed for buckwheat pasta produced with 30% of flour moisture and 100 rpm of the extruder screw”. However, the data in Table 2 are not consistent with this description. Please explain this!

6)    Page 1, line 20. “was” should be changed to “were”

7)    Page 2, line 46. “functional food” should be changed to “functional foods”

8)    Page 2, line 67. “process” should be changed to “processes”

9)    Page 2, line 71. “was” should be changed to “is”, and this sentence should be indented for two characters.

10) Page 3, line 108. “Out” should be changed to “Our”

11) Page 3, line 125. “depend to” should be changed to “depend on”

12) Page 5, line 177. “each analyzed phenolic acids” should be changed to “each analyzed phenolic acid”

13) Page 7, line 206. I don’t understand what “The extraction proper” means?

14) The format of references must be carefully checked. For example, reference 1 “Xingqiang, W.; Xusheng, G.; Shuxuan, L.; Yunkai, L.; Hanwen, S.”; reference 6 “Wang, M.; Liu, J.-R.; Gao, J.-M.; Parry, J.W.; Wei, Y.-M.”; reference 12 and so on.

Author Response

Dear Reviewer,

The authors would like to thank the Reviewer for your valuable comments. The text has been corrected according to suggestions of the Reviewer. . Manuscript have been checked by a native English speaking person from “Two Heads. Coaching and Language” company. The proper changes have been introduced.

Yours sincerely,

Anna Oniszczuk

1) In the whole manuscript, the mechanism of the effects of processing parameters on the content of phenolic compounds is not analyzed and discussed.

The required information has been now added to the text (lines 123-138).

2)    This research received no external funding, just in my opinion, I consider that this study is not a scientific research in the current situation, but a technical study.

This study was supported by funds of Medical University of Lublin granted by Polish Ministry of Science and Higher Education.

3)    HPLC-ESI-MS/MS means “high-performance liquid chromatography electrospray ionization tandem mass spectrometry” rather than “high performance liquid chromatography coupled with mass spectrometry”, the authors should revise it through the whole manuscript.

The proper changes have been now made in the text.

4)    A total of 15 phenolic acids described, why only 11 phenolic acids presented in Table 2?

Fifteen phenolic acids were detected in pasta samples. They were: gallic, protocatechuic, gentisic, 4- hydroxybenzoic, vanilic, trans-caffeic, cis-caffeic, trans-p-coumaric, cis-p-coumaric, syryngic, trans-ferulic, cis-ferulic, salicylic, trans-sinapic and cis-sinapic. The presence of syryngic, trans-ferulic and sinapic acids was lower than the limit of quantification but higher than the limit of detection. They could not be quantified, therefore they are not presented in Table 2.

5)    Page 3, line 94-96. The authors said that “However, the highest content of cinnamic acid derivatives trans-caffeic, trans-p-coumaric, cis-p-coumaric and cis-ferulic was observed for buckwheat pasta produced with 30% of flour moisture and 100 rpm of the extruder screw”. However, the data in Table 2 are not consistent with this description. Please explain this!

I apologize for my typing error. The highest content of cinnamic acid derivatives trans-caffeic, trans-p-coumaric, cis-p-coumaric and cis-ferulic was observed for buckwheat pasta produced with 30% of flour moisture and 60 rpm of the extruder screw. It is now corrected in the manuscript.

Ad. 6)-14) The proper changes have been now made in the text.

6)    Page 1, line 20. “was” should be changed to “were”

7)    Page 2, line 46. “functional food” should be changed to “functional foods”

8)    Page 2, line 67. “process” should be changed to “processes” – it has been changed for high-temperature/high-pressure process because extrusion-cooking treatment is one complexed processing method

9)    Page 2, line 71. “was” should be changed to “is”, and this sentence should be indented for two characters.

10) Page 3, line 108. “Out” should be changed to “Our”

11) Page 3, line 125. “depend to” should be changed to “depend on”

12) Page 5, line 177. “each analyzed phenolic acids” should be changed to “each analyzed phenolic acid”

13) Page 7, line 206. I don’t understand what “The extraction proper” means?

14) The format of references must be carefully checked. For example, reference 1 “Xingqiang, W.; Xusheng, G.; Shuxuan, L.; Yunkai, L.; Hanwen, S.”; reference 6 “Wang, M.; Liu, J.-R.; Gao, J.-M.; Parry, J.W.; Wei, Y.-M.”; reference 12 and so on.

Reviewer 3 Report

The paper is about the effects of production parameters on the content of phenolic acids in extruded pasta products made from flour from roasted buckwheat grain. The paper is interesting and gives new information about impact of extrusion parameters on the composition of phenolic acids. However, there are some critical points that need to be clarified or corrected before publication:

Table 1. please edit the Table in order to make shortly 

Section 2.1 Have the authors used an internal standart or QC?. How they controled the efficiency of the equipment?  

Line 81. The authors are sure that an elevate temperature as 60ºC could not affect the integrity of polyphenols.

Line 116. Please to remove trance-p-coumaric and write trans-p-coumaric.

Line 144 Please when the acronym is used at the first time it is convenient to explain the significance TLC-DPPH. The same observation is to Line 148, DPPH.

Table 3 Please to write relation instead relatio.

Line 191 Please to indicate the meaning of AACC

Line 202. Did The authors make a reextraction in the preparation of polyphenolic extracts? It would be interesting because this procedure makes sure that the totally of polyphenolic content was completely extracted. Did they used internal standart in ordert to evaluate the efficiency of the extraction method?. Has this method been validated?

Author Response

Dear Reviewer,

The authors would like to thank the Reviewer for your valuable comments. The text has been corrected according to suggestions of the Reviewer. . Manuscript have been checked by a native English speaking person from “Two Heads. Coaching and Language” company. The proper changes have been introduced.

Yours sincerely,

Anna Oniszczuk

The paper is about the effects of production parameters on the content of phenolic acids in extruded pasta products made from flour from roasted buckwheat grain. The paper is interesting and gives new information about impact of extrusion parameters on the composition of phenolic acids. However, there are some critical points that need to be clarified or corrected before publication:

Table 1. please edit the Table in order to make shortly

Proper changes have been now made in the Table 1.

Section 2.1 Have the authors used an internal standart or QC? How they controled the efficiency of the equipment? 

We have used external standards for optimization of ESI-MS/MS parameters and quantitation.

QC sample with all standard compounds at known concentrations was run every 10th sample to control instrument response and retention times. No fluctuations of retention times or in instrument response were observed.

Line 81. The authors are sure that an elevate temperature as 60ºC could not affect the integrity of polyphenols.

An extremely significant step of sample pretreatment is the choice of extraction conditions. Therefore, before quantitative analysis, optimization of ultrasound assisted extraction of phenolic acids from precooked buckwheat pasta was performed. Optimization of extraction solvent was the first step of the experiment. In order to achieve this, 80 % aqueous ethanol and 80 % aqueous methanol were used. Eighty percent of EtOH appears to be the superior extractant for the isolation of analyzed compounds from buckwheat pasta. Moreover, this solvent meets the requirements of Green Chemistry. The next step was optimization of the extraction temperature. Extraction temperature was adjusted to 55, 60, and 65 °C. The increase of temperature from 55 to 60 °C improves efficiency of the process, whereas the further increase of temperature reduces the efficiency of extraction. It can be supposed that temperature higher than  60 °C (and cavitation in these conditions) probably caused the degradation of phenolic acids.

Lines: 116, 144, 148, 191, Table 3 - the proper changes have been now made in the text.

Line 116. Please to remove trance-p-coumaric and write trans-p-coumaric.

Line 144 Please when the acronym is used at the first time it is convenient to explain the significance TLC-DPPH. The same observation is to Line 148, DPPH.

Table 3 Please to write relation instead relatio.

Line 191 Please to indicate the meaning of AACC

Line 202. Did The authors make a reextraction in the preparation of polyphenolic extracts? It would be interesting because this procedure makes sure that the totally of polyphenolic content was completely extracted. Did they used internal standart in ordert to evaluate the efficiency of the extraction method?. Has this method been validated?

For the first extraction, 2 g of sample was soaked in 40 mL of 80% ethanol and extracted  in an ultrasonic bath at 60°C for 20 minute. The remnant from the first extraction was used as the raw material for the re-extraction process. It was soaked 80% ethanol and extracted by the same way as mentioned above. The obtained extracts were  filtered, combined and evaporated to a total volume 5 ml.

The method was validated. The accuracy of the extraction method was evaluated through recovery studies by adding already known amounts of the each standard solution (three concentration levels). The recoveries were in the range of 91.89 for protocatechuic acid to 98.23% for vanilic acid.

Round 2

Reviewer 2 Report

The format of reference needs further normalization. The typical MRM chromatograhy diagrams for analyzed compounds should be provided in supplementary materials.

Author Response

Dear Reviewer,

I am pleased to resubmit for publication the revised version of my manuscript. I appreciate your kindness, efforts and comments.

The authors would like to thank the Reviewer for valuable comments which have helped to improve the quality of the manuscript. Proper changes and now made.

1.      The format of reference needs further normalization.

The References have been corrected according to suggestions of the Reviewer.

2.      The typical MRM chromatograhy diagrams for analyzed compounds should be provided in supplementary materials.

Exemplary LC-MS/MS chromatogram of phenolic acids from precooked buckwheat pasta is now provided.

Kind regards,

Anna Oniszczuk